

# SAMGAT: structure-aware multilevel graph attention networks for automatic rumor detection

Yafang Li[1], Zhihua Chu[1], Caiyan Jia[2] and Baokai Zu[1]

[1] Faculty of lnformation Technology, Beijing University of Technology, Beijing, China
[2] School of Computer and Information Technology & Beijing Key Laboratory of Traffic Data Analysis and Mining, Beijing Jiaotong University, Beijing, China

## ABSTRACT

The rapid dissemination of unverified information through social platforms like Twitter poses considerable dangers to societal stability. Identifying real versus fake claims is challenging, and previous work on rumor detection methods often fails to effectively capture propagation structure features. These methods also often overlook the presence of comments irrelevant to the discussion topic of the source post. To address this, we introduce a novel approach: the Structure-Aware Multilevel Graph Attention Network (SAMGAT) for rumor classification. SAMGAT employs a dynamic attention mechanism that blends GATv2 and dot-product attention to capture the contextual relationships between posts, allowing for varying attention scores based on the stance of the central node. The model incorporates a structure-aware attention mechanism that learns attention weights that can indicate the existence of edges, effectively reflecting the propagation structure of rumors. Moreover, SAMGAT incorporates a top-k attention filtering mechanism to select the most relevant neighboring nodes, enhancing its ability to focus on the key structural features of rumor propagation. Furthermore, SAMGAT includes a claim-guided attention pooling mechanism with a thresholding step to focus on the most informative posts when constructing the event representation. Experimental results on benchmark datasets demonstrate that SAMGAT outperforms state-of-the-art methods in identifying rumors and improves the effectiveness of early rumor detection.

## INTRODUCTION

The widespread adoption of social media has decentralized the authority over information, leading to scenarios where truth is no longer determined solely by authoritative sources. This shift has significant implications for society as misinformation can profoundly affect political stability, public health, and social cohesion. For instance, during political elections, misinformation campaigns have been known to manipulate public opinion, thereby undermining democratic processes. Similarly, during the COVID-19 pandemic, the spread of false information regarding treatments and the severity of the virus has led to

Corresponding author
Baokai Zu, zubaokai@163.com,
bzu@bjut.edu.cn

public confusion, non-compliance with health guidelines, and vaccine hesitancy, further exacerbating public health crises. Therefore, the need for effective rumor detection methods is more urgent than ever. Deep learning has made a major impact on rumor detection, capable of extracting deep semantic information from various data types, including text, images, and structural information of rumors (*Li, Zhang & Si, 2019*). Traditional methods, such as *Recurrent Neural Networks* (RNNs) and *Convolutional Neural Networks* (CNNs), have shown effectiveness in capturing time series relationships and local spatial feature representation respectively (*Ma et al., 2016*; *Wu et al., 2020*; *Lu & Li, 2020*; *Yu et al., 2017b*; *Kumar, Bhatia & Sangwan, 2022*). However, these methods are limited to using just text information, often overlooking the structural information of rumor propagation. As a result, several studies have investigated incorporating details about how rumors spread into rumor detection models by utilizing approaches grounded in *Graph Neural Network* (GNNs) (*Bian et al., 2020*; *Lu & Li, 2020*; *Nguyen et al., 2020*; *Yuan et al., 2019*; *Yu et al., 2022*). However, GNNs often struggle to learn optimal graph representations due to the presence of unrelated or irrelevant connections.

Given the complexity and diversity of social media content, rumor detection in social networks is challenging. Misinformation is often unconsciously generated and spread by regular users, while malicious actors design confusing dialogue structures to propagate rumors. These factors contribute to the noise and complexity in the data, making it difficult for existing models to accurately detect rumors. Furthermore, existing methods face several limitations in effectively extracting structural information. Many current approaches consider all nodes within the graph without filtering out irrelevant ones, leading to the inclusion of noisy and unrelated information. Additionally, they often lack explicit supervision in learning propagation structures, relying solely on the inherent graph structure without leveraging prior knowledge to guide the learning process. Moreover, the modeling of pairwise relationships remains inefficient, as traditional attention mechanisms struggle to capture the nuanced importance of various connections within the graph. These challenges and limitations highlight the need for an improved method of discerning the importance of various relations within the graph to enhance the effectiveness of rumor detection.

Graph attention networks (GATs) (*Veličković et al., 2018*) have been utilized to address these challenges, employing self-attention to capture node importance. Despite their utility, GATs are not without limitations, particularly when nodes with highly prominent features disproportionately affect attention scores. This can lead to skewed representations, analogous to scenarios in rumor detection where a few "hot" comments might dominate and thus potentially mislead the model's judgments. To address this imbalance, *Min et al. (2022)* and *Zhang et al. (2023)* implemented an enhancement by integrating GATv2 (*Brody, Alon & Yahav, 2022*). Taking this innovation further, we have developed DynamicGAT, which incorporates a universal approximation attention function and DP attention. This enhancement ensures dynamic and robust attention, enabling the model to incorporate a wider range of connections and mitigate the risk of being misled by overly influential nodes. The DP attention mechanism utilizes a sigmoid function to represent the connection strength, refining the model's focus on the relevance of connections between nodes. This

integration enables the model to discern and prioritize the most meaningful connections, thus avoiding distraction from irrelevant information. In summary, we propose a robust graph attention model, DynamicGAT, that addresses the complexities of social media for enhanced rumor detection.

To leverage propagation structure, we enhance the relational importance within graphs *via* self-supervised attention. This leverages edges encoding relation importance. If nodes $i$ and $j$ are linked and similar in content, they are deemed relevant to each other; if not, they're less important. Traditional attention mechanisms lack direct supervision. In contrast, our approach exploits prior knowledge to guide attention (*Knyazev, Taylor & Amer, 2019*; *Yu et al., 2017b*). This reinforcement not only elevates the significance of authentic connections but also mitigates the impact of forged or unrelated links, thus enabling the model to deliver robust predictions even in the face of perturbed graph structures.

In rumor detection, the number of replies to a post is a beneficial indicator (*Wu, Yang & Zhu, 2015*). Our model maintains this cardinality information, denoting the number of reposts, while managing complex and noisy post connections. We propose a dual mechanism where attention is first concentrated on the top-k most similar nodes to the central node. Additionally, to avoid loss of cardinality information, our model assigns uniform attention weights to all neighboring nodes, incorporating this into the original attention-based representation. We further advance existing graph attention pooling method to form constrained, claim-guided graph representation. This approach retains the overall graph structure, refines node representations, and prioritizes more relevant posts. Thus, we balance maintaining cardinality and refining the graph for effective rumor detection within a single, robust model.

To summarize, the key contributions of the present work are:

- We put forth an innovative constrained dynamic graph attention model to effectively prioritize critical posts, representing events more accurately while preserving cardinality information and modeling different relationships simultaneously.
- We guide graph attention using conversation structure in a self-supervised manner, outperforming traditional graph attention models. Our model leverages trustworthy prior structural knowledge to provide robust performance even under perturbed graph structures.
- We restrict posts in the graph pooling process based on constrained claim-guided attention, demonstrating superior performance in rumor detection.

## RELATED WORK

The rampant spread of misinformation on social media platforms has made the development of effective rumor detection methods a crucial endeavor. Traditional machine learning classifiers have laid the groundwork for rumor detection by utilizing handcrafted features such as sentiment analysis, bag-of-words models, user-profile features, text style and temporal patterns (*Castillo, Mendoza & Poblete, 2011*; *Ma et al., 2015*; *Tolosi, Tagarev & Georgiev, 2016*; *Enayet & El-Beltagy, 2017*; *Kumar, Bhatia & Sangwan, 2022*). These

approaches, while pioneering, were limited by their reliance on surface features and a lack of depth in capturing the complex nature of misinformation spread.

The advent of deep learning has significantly advanced the field of rumor detection. The introduction of RNNs by *Ma et al. (2016)* marked a significant shift toward models capable of capturing temporal semantic information, superseding models dependent on handcrafted features. Subsequent work enhanced LSTM with attention mechanisms to refine text representations and model temporal relationships within posts (*Chen et al., 2018*). *Yu et al. (2017a)* groups posts into temporal windows, extracting representations using paragraph vectors for each group and subsequently utilizing CNNs to model local patterns. Similarly, *Chen et al. (2019)* employed a grouping method but further enhanced group representations using attention mechanisms, thereby improving the capture of global features. The PLAN model (*Khoo et al., 2020*) further advanced global feature extraction by employing Transformer networks to model pairwise interactions between tweets through self-attention.

Despite these advancements in capturing temporal, local, and global representations of rumors, most approaches have not fully considered the complex propagation structures inherent to rumors. Addressing this gap, several studies have pivoted towards the structural characteristics of rumor spreading. *Ma, Gao & Wong (2018)* introduced recursive neural networks that effectively modeled the bottom-up and top-down propagation trees. Subsequently, graph convolutional networks (GCNs) were utilized to represent both propagation and dispersion graphs (*Bian et al., 2020*). *He et al. (2021)* improved the work (*Bian et al., 2020*) by using event-level contrastive learning. Further developments by *Yuan et al. (2019)* as well as *Lu & Li (2020)* conceptualized propagation structures as graphs incorporating heterogeneous information networks. EBGCN (*Wei et al., 2021*) addressed uncertainties in propagation structures using a bayesian method to adjust weights of unreliable relations. HD-Trans (*Ma & Gao, 2020*) refined the PLAN approach by restricting self-attention to interactions within subtrees, thereby capturing hierarchical relationships. This method can be seen as a bidirectional graph attention network (GAT) incorporating sibling edges, employing Transformer-style attention functions. During graph pooling, it selectively focuses on the most relevant information through attention mechanisms to aggregate a comprehensive event representation. Similar to HD-Trans, the model proposed by *Lin et al. (2021)* learns attentional representations at the post level, incorporating both event-level attention and sibling relationships. However, it uniquely conditions its event-level attention specifically on the source post, setting it apart from the HD-Trans approach.

Recently, more studies have considered fusing multiple features for rumor detection. DynamicGCN (*Choi et al., 2021*) represents a step forward in rumor detection by integrating temporal dynamics and structural information through the use of dynamic graph convolutional networks with attention mechanisms. TISN (*Luo et al., 2022*) uses GCN to capture propagation patterns and concatenates posts in chronological order, employing transformers to extract temporal features. UMLARD (*Chen et al., 2022*) integrates user profiles, structural and temporal features, and tweet content through

multi-view learning and attention mechanisms for enhanced rumor detection. DAN-Tree++ (*Bai, Han & Jia, 2023*) enhances the PLAN model by integrating user profile data and introducing tree structure embedding into Transformer blocks. MSLG (*Han et al., 2023*) summarizes multi-source information into a heterogeneous graph that includes rumor-word, word-word, and rumor-user relationships. Despite considering a broader scope of information, existing approaches still face limitations in effectively extracting structural information. Notably, current methods often consider all nodes within the graph without filtering out irrelevant ones, and they lack explicit supervision in learning propagation structures. Moreover, the modeling of pairwise relationships remains inefficient. To address these limitations, we propose Structure-Aware Multilevel Graph Attention Networks (SAMGAT), an extension of the GAT designed to enhance the modeling of propagation structures.

# METHODOLOGY

This section presents our SAMGAT for rumor detection leveraging undirected graph structure. Our proposed model aggregates information using graph attention operating at different levels, *i.e.,* a graph attention mechanism analyzes the local context by capturing the importance of neighboring tweets for each node, while a claim-guided attention mechanism leverages global information from a central "claim" node, enhancing the understanding of post content within the broader context of the interaction graph. As shown in Fig. 1, our model will be further described in the subsequent subsections.

## Problem definition

We define the dataset for rumor detection as $C = \{c_1, c_2, \ldots, c_m\}$, with each $c_i$ corresponding to a distinct rumor event. $c_i = \left\{r_i, \omega_1^i, \omega_2^i, \ldots, \omega_{n_i-1}^i, G_i\right\}$, where $n_i$ denotes the post counts in that event, $r_i$ is the claim post (*Ma, Gao & Wong, 2018*), $\omega_j^i$ is the $j$th responsive post, and $G_i$ symbolizes the graph structure. In particular, we utilize $G_i$ to represent a graph consisting of the node set $V_i$ which represents posts in that event and the edge set $E_i$ which denotes the responsive relationships between posts. The graph node feature of $G_i$ is represented as $X_i = \left[x_0^i, x_1^i, \ldots, x_{n_i-1}^i\right]$, where $x_j^i \in \mathbb{R}^d$ denotes the word embedding of each post. Let $A_i$ be the adjacency matrix of $G_i$.

Additionally, the goal of rumor detection is to learn a classifier $f : C \rightarrow Y$ that maps events $C$ to their labels $Y$. Specifically, each event's label can take one of two values: either FR for false rumor or TR for true rumor. In some datasets, labels are classified with greater specificity, taking one of four values from the set NR, FR, TR, UR. These finer-grained labels represent: non-rumor, false rumor, true rumor, and unverified rumor. For clarity, we have summarized some of the key terms and notations in Table 1.

## Graph attention networks

GATs pivot on enriching the representation of posts by assigning differential importance to neighboring posts, rather than homogenizing their importance as in the case of the GCN model. The rationale behind employing GATs for embedding interaction graphs is to mitigate the impact of unrelated posts.

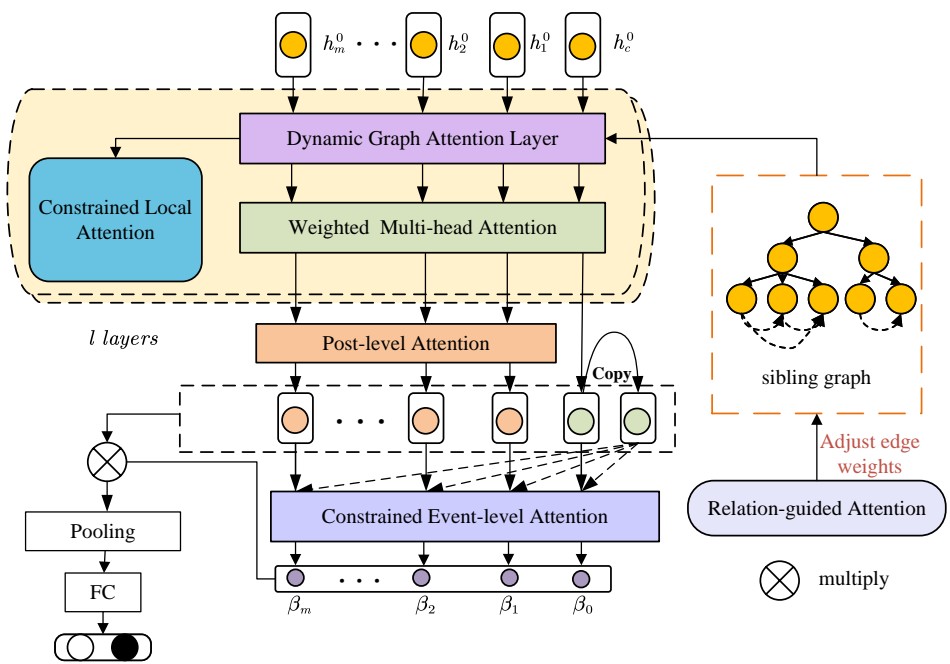

**Figure 1** **The framework of our proposed structure-aware multilevel graph attention networks.** This architecture integrates several key components built upon the Sibling Graph that represents semantic relationships. The graph attention layer computes attention scores between nodes, leveraging the structured information, as depicted in Fig. 2. The constrained local attention imposes limits on attention weight, elaborated in Fig. 3. Furthermore, the relation-guided attention component adjusts edge weights to enhance feature representation, depicted in Fig. 4. Our network efficiently encodes the graph topology and node interrelations through these interconnected modules, coupled with the Post-level and Constrained Event-level Attention.

Leveraging self-attention in the form of GAT marks our starting point. Our goal is to aggregate similar viewpoints of different types among neighboring nodes. Consequently, the attention weights reflect the influence of neighboring posts on the focal post within the graph $G$. The latent representations of nodes at layer $l$ is denoted by $H^l = \left[ h_c^l, h_1^l, h_2^l, \ldots, h_{|V|-1}^l \right]^\top$. Here, $h_c^l$ is equivalently represented as $h_0^l$. Initially, $H^0 = X$. The computation of the attention weight proceeds as follows:

$$\alpha_{ij}^l = \text{Atten}\left( h_i^l, h_j^l \right) \tag{1}$$

$$e_{ij}^l = LeakyReLU\left( \vec{a}^\top \left[ W^l h_i^l \parallel W^l h_j^l \right] \right) \tag{2}$$

$$\alpha_{ij}^l = \frac{\exp\left( e_{ij}^l \right)}{\sum_{k \in \mathcal{N}_i} \exp\left( e_{ik}^l \right)} \tag{3}$$

In this context, the function Atten computes the attention mechanism. Here, $h_i^l$ and $h_j^l$ represent the hidden states of tweets $\omega_i$ and $\omega_j$ in the graph, respectively. $\alpha_{ij}^l$ reflects the

**Table 1  Terms and notations used in the article.**

| Symbol | Definition |
|---|---|
| $X$ | Features of $N$ nodes. |
| $A$ | Adjacency matrix of a graph. |
| $\alpha_{ij}^l$ | Attention score between posts $i$ and $j$ at layer $l$. |
| $e_{ij}^l$ | Attention score before softmax between posts $i$ and $j$ at layer $l$. |
| $\mathcal{N}_i$ | One-hop neighbors of node $i$. |
| $h_i^l$ | Hidden state of node $i$ at layer $l$. |
| $e_{ij,\text{GO}}^l$ | GAT Original attention score. |
| $e_{ij,\text{DP}}^l$ | Dot-product attention score. |
| $e_{ij,\text{MX}}$ | MX attention score combining GO and DP. |
| $\tilde{\alpha}_{ij}$ | Modified attention weight after constrained local attention. |
| $\phi_{ij}$ | Probability of connection between nodes $i$ and $j$. |
| $g_{c \rightarrow i}^L$ | Gate vector for post-level attention. |
| $\bar{h}_i^L$ | Modified hidden representation after post-level attention. |
| $h_i^c$ | Joint representation of node $i$ and the claim. |

relevance of tweet $\omega_j$ to $\omega_i$, while $\vec{a}$ parameterize a weight vector to computes attention weights and $W^l$ represents a trainable transformation matrix specific to each layer. Here, $\cdot^\top$ represents transposition and the $\parallel$ symbol denotes the "concatenate" operation, and $\mathcal{N}_i$ includes $\omega_i$'s one-hop neighbors along with $\omega_i$ itself. The activation function is the LeakyReLU. This leads us to the definition of the graph attention layer:

$$h_i^{l+1} = ReLU\left(\sum_{j \in \mathcal{N}_i} \alpha_{ij}^l W^l h_j^l\right) \tag{4}$$

### Weighted multi-head attention

To understand and represent different types of responsive relationships, we introduce multi-head attention (*Vaswani et al., 2017*) in Eq. (5) by averaging the outputs from $K$ attention heads:

$$h_i^{l+1} = ReLU\left(\frac{1}{K}\sum_{k=1}^{K}\sum_{j \in \mathcal{N}_i} \alpha_{ij}^{lk} W_k^l h_{jk}^l\right) \tag{5}$$

where $h_i^{l+1}$ signifies the latent representations of the post $\omega_i$ in the $(l+1)$-th layer. $\alpha_{ij}^{lk}$ indicates the normalized attention weight determined by the $k$-th head in the $l$-th layer, and $W_k^l$ refers to the learnable parameters associated with the linear transformation at that layer. The resulting embedding for the $l+1$ layer is obtained through an averaging process.

After going through $L-1$ layers of GAT, we propose weighted multi-head attention to manage the contributions of different relationships. Equation (5) is further enhanced by Eq. (6) *via* weighting different relationships. This extension allows our model to differentiate and prioritize diverse perspectives and attitudes in user comments, building

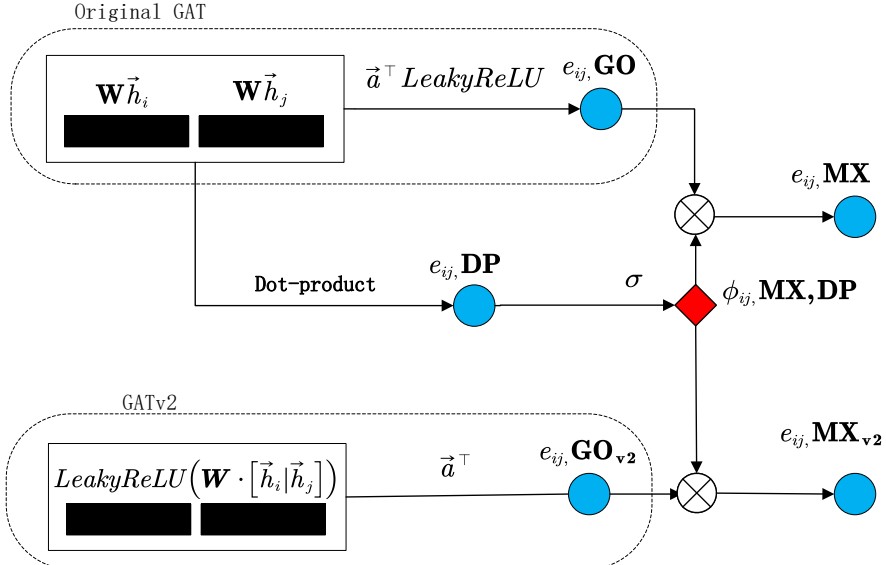

**Figure 2 Overview of graph attention layer: GO, DP, MX, GO$_{v2}$, MX$_{v2}$.** The unnormalized attention before softmax is represented by blue circles ($e_{ij}$), while the normalized attention between node $i$ and $j$ is denoted by red diamonds ($\phi_{ij}$).

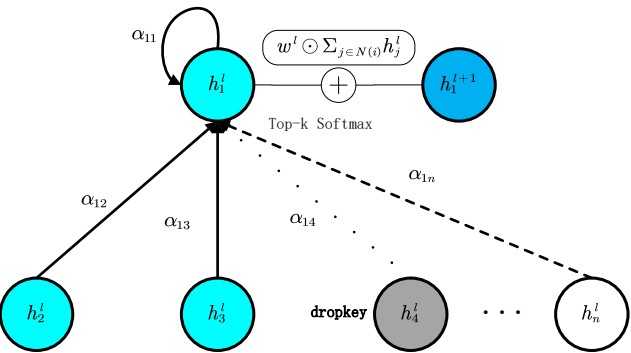

**Figure 3 In the process of constrained local attention, the dropped key nodes for node *i* are indicated by grey circles.** The top-k nodes are represented by blue circles, while the nodes beyond the top-k that are excluded from attention are shown as white circles. $W^l$ is omitted for simplicity. $\odot$ denotes the element-wise multiplication.

upon the foundation laid by the GAT. The computation of the final output in the $L$-th layer is calculated using a newly introduced weight vector $c$.

$$h_i^L = ReLU\left(\sum_{k=1}^{K} c_k \sum_{j \in \mathcal{N}_i} \alpha_{ij}^{L-1,k} W_k^{L-1} h_j^{L-1}\right) \tag{6}$$

$h_i^L$ represents the enhanced node representation of $\omega_i$, which encodes the weighted contributions of different relationships. We utilize mean pooling to summarize the whole

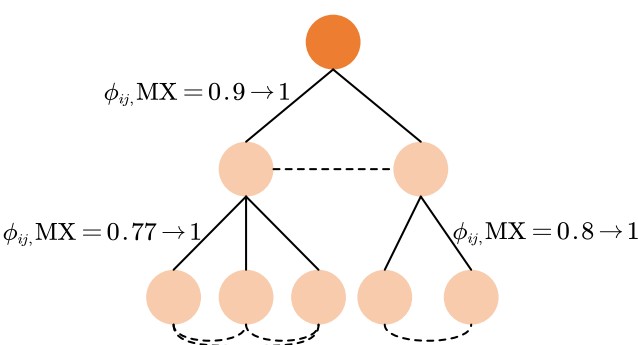

**Figure 4** **The process of relation guided attention.** The sibling graph is formulated by augmenting the graph with additional edges connecting any pairs of nodes that share the same parent node. The target attention weight between directly connected nodes is set to 1, while the target attention weight between sibling nodes is parameterized by a hyperparameter $\gamma^s$ ranging between 0 and 1.

event.

$$\bar{s} = meanpooling\left(H^L\right) \tag{7}$$

Here, $\bar{s}$ refers to the representation of the entire graph obtained through mean-pooling.

### DynamicGAT

Building upon the GAT, we introduce DynamicGAT, which includes two variations of the attention mechanism, namely GAT Original (GO) and dot-product (DP). These forms of attention aim to further refine our model, allowing it to better understand and represent data:

$$e^l_{ij,\text{GO}} = LeakyReLU\left(\vec{a}^\top\left[W^l h^l_i \| W^l h^l_j\right]\right) = LeakyReLU\left(\vec{a}_1^\top W^l h^l_i + \vec{a}_2^\top W^l h^l_j\right)$$
$$= LeakyReLU\left(W^l_1 h^l_i + W^l_2 h^l_j\right) \tag{8}$$

where $\vec{a}_1 \| \vec{a}_2 = \vec{a}$, and $\vec{a}_1^\top W^l$ and $\vec{a}_2^\top W^l$ can be collapsed into two single linear transformations $W^l_1$ and $W^l_2$, respectively, and

$$e^l_{ij,\text{DP}} = \left(W^l h^l_i\right)^\top \cdot \left(W^l h^l_j\right) \tag{9}$$

Building on the enhanced capabilities of the GATv2 model (denoted as $GO_{v2}$) (*Brody, Alon & Yahav, 2022*), our approach incorporates a shared parameter weight $W$ and instigates nonlinearity through the placement of the LeakyReLU function before $\vec{a}^T$. This modification enables GATv2 to achieve a universal approximation attention function, thereby endowing it with greater expressive power than its predecessor, GAT. The original GAT model operates under a static attention mechanism, where the ranking of attention scores among neighboring nodes remains constant across different central nodes. This means that regardless of whether the central node represents support or opposition, GAT assigns the highest attention to the same neighbor node—presumably the one with the highest support. Ideally, the model should aggregate neighbors that align with the central

node's stance. In contrast, GATv2′s formulation takes into account the relationship between the central node and its neighbors, allowing for a dynamic attention mechanism that can vary the attention scores based on the context provided by the central node:

$$e^l_{ij,\text{GO}_{v2}} = \vec{a}^\top LeakyReLU\left(W^l h^l_i + W^l h^l_j\right) \tag{10}$$

Moreover, the ultimate attention mechanism of DynamicGAT, denoted as MX, is obtained by blending the GO and DP types. Specifically, MX is calculated by element-wise multiplication of $\text{GO}_{v2}$ and DP attention values followed by a sigmoid activation function $\sigma$. In the field of natural language inference (NLI), DP attention mechanisms have been widely employed to assess the compatibility between two sentences (*Hu et al., 2020*). Since DP attention, combined with the sigmoid function, represents the degree of agreement, it can subtly weaken the influence of nodes that contradict the view while implicitly emphasizing those that share a common perspective.

$$\text{DynamicGAT} : e_{ij,\text{MX}_{v2}} = e_{ij,\text{GO}_{v2}} \cdot \sigma\left(e_{ij,\text{DP}}\right) \tag{11}$$

### Constrained local attention

Experiments have confirmed that attention dropout is helpful for GAT training on small graphs. *Fang et al. (2023)* proves that integrating non-biased random dropping strategies into graph neural networks equates to adding an extra term for regularization, which leads to a more robust model. Taking inspiration from the dropkey technique utilized in the Swin-transformer (*Li et al., 2023*), we prefer to use dropkey rather than traditional dropout when computing the final attention scores. We specifically introduce a novel dropout-before-softmax scheme where the Key is set as the dropout unit. Importantly, we generate a unique masked key map for each central node instead of sharing a single map across all query vectors. This scheme can regularize attention weights and keep them as a probability distribution simultaneously.

$$d_{ij} = \begin{cases} 0 & \text{with probability } 1-d \\ -\infty & \text{with probability } d \end{cases} \tag{12}$$

$$\alpha_{ij} = \frac{\exp\left(d_{ij} + e_{ij}\right)}{\sum_{k\in\mathcal{N}(i)}\exp\left(d_{ik} + e_{ik}\right)}. \tag{13}$$

Furthermore, in order to control the aggregation of information and eliminate the potential impact of noisy edges, we impose a restriction on the number of nodes $j$ that can be aggregated for each central node $i$, using a hyperparameter $p_a$. By selecting only the top $p_a$ nodes with the highest attention weights to aggregate, we effectively filter out the nodes that are least relevant to the central node, which are more likely to be connected by noisy edges. This is done to ensure that a post retains its own information and to prevent the aggregation of irrelevant or noisy information from neighboring nodes, which could degrade the performance of rumor detection.

$$\tilde{\alpha}_i = \text{TopK}\left(\alpha_i, p_a\right) \tag{14}$$

Graph structure can be difficult to distinguish for attention-based graph neural networks. Since rumor detection is formulated as a graph classification task, it is necessary to enhance the discriminative ability of our GNN. In light of the limitations of attention-based aggregators in preserving the cardinality of multisets (*Zhang & Xie, 2020*), we introduce modifications to the weighted summation function to address this issue. Specifically, we incorporate cardinality information into the aggregation process without altering the attention function. This approach ensures that each node in the neighborhood contributes to the central node, implicitly maintaining the cardinality information. By doing so, we not only keep the expressive power of the DynamicGAT but also improve the effectiveness of node feature representation, thereby strengthening the discriminative capabilities of the GNN. The modified equation is presented below:

$$h_i^{l+1} = ReLU \left( \sum_{j \in \mathcal{N}(i)} \tilde{\alpha}_{ij}^l W^l h_j^l + w^l \sum_{j \in \mathcal{N}(i)} W^l h_j^l \right) \tag{15}$$

Here, $w^l$ is the weight vector that is multiplied with the node features to adjust the influence of the cardinality of the multiset.

### Relation-guided attention

Inspired by SuperGAT (*Kim & Oh, 2021*), we can leverage the presence of edges to supervise graph attention. This approach is grounded in the label-agreement assumption, which posits that the relationship between connected nodes should be closer than that between unconnected nodes. We leverage the link prediction task to provide self-supervision for attention, using labels derived from the presence or absence of edges: for a pair of nodes $i$ and $j$, the label is 1 if an edge exists between them and 0 otherwise. We also consider sibling relationships and label them as $y^s$ between 0 and 1. This mechanism, while being a part of our GAT-based model, provides additional guidance to our attention mechanism. It can be viewed as learning the graph structure because the learned node representations can reconstruct both the reply relationships between posts and the sibling relationships of posts. Hence, we refer to it as structure-aware attention. We sample a ratio of sibling edges and set this ratio as a hyperparameter $\eta$. We use DP attention with sigmoid $\sigma$ to calculate the probability $\phi_{ij}$ of connection between node $i$ and $j$. Note that the computation of DP attention only requires the representations of the two nodes. Therefore, we are essentially learning node representations that can reflect the graph structure.

$$\phi_{ij,\mathrm{MX}} = \sigma \left( e_{ij,\mathrm{DP}} \right) \tag{16}$$

Training samples consist of three parts: connected edges $E$, sibling edge samples $E^s$, and the complementary set $E^c = (V \times V) \setminus (E \cup E^s)$. However, for graphs containing a vast number of nodes, utilizing every possible negative case in $E^c$ would not be computationally efficient. Not all connected and sibling posts are valid, because there might be bot-generated content. Therefore, we first arbitrarily choose a total of $p_n \cdot |E|$ negative samples $E^-$ from $E^c$ where the negative sampling ratio $p_n \in \mathbb{R}^+$ is a hyperparameter. After sampling edges, we calculate their corresponding nodes' attention weight by dot-product attention, and select the top $r$ ratio of edges in ascending or descending order as $\hat{E}$, $E^s$ and $\hat{E}^-$. Due to the

possibility of implicit connections between unconnected posts, we select those that are less likely to be connected to serve as negative edges. The top $r$ ratios for each edge type are set differently to balance the importance of each type in the training process.

$$\hat{E} = \text{sort}(E, ratio = r_c, \text{key} = \phi_{ij}, \text{ascending} = \textit{False}) \tag{17}$$

$$\hat{E}^s = \text{sort}(E^s, ratio = r_s, \text{key} = \phi_{ij}, \text{ascending} = \textit{False}) \tag{18}$$

$$\hat{E}^- = \text{sort}(E^-, ratio = r_n, \text{key} = \phi_{ij}, \text{ascending} = \textit{True}) \tag{19}$$

where $r_c$, $r_s$, and $r_n$ are the top $r$ ratios for connected edges, sibling edges, and negative edges, respectively. These ratios are hyperparameters that can be tuned based on the specific task and dataset characteristics to achieve the best performance.

The concept behind this approach is akin to SGATs (*Ye & Ji, 2023*), which is capable of identifying and eliminating task-irrelevant edges in graphs, thereby producing robust outcomes even in the presence of noisy graphs. We apply a similar concept to ensure the effectiveness of SuperGAT in noisy graph environments by exclusively utilizing reliable edges. The loss function $\mathcal{L}_E^l$ of layer $l$ is:

$$\mathcal{L}_E^l = - \left( \begin{array}{c} \dfrac{1}{|\hat{E} \cup \hat{E}^-|} \displaystyle\sum_{(j,i) \in \hat{E} \cup \hat{E}^-} 1_{(j,i)=1} \cdot \log\phi_{ij}^l + 1_{(j,i)=0} \cdot \log\left(1 - \phi_{ij}^l\right) \\ + \dfrac{1}{|\hat{E}^s|} \displaystyle\sum_{(j,i) \in \hat{E}^s} \left| y^s - \phi_{ij}^l \right|^2 \end{array} \right) \tag{20}$$

where $1_{(j,i)=1}$ and $1_{(j,i)=0}$ are indicator functions. An indicator function is a common mathematical concept used to determine whether a certain condition is true or false. These indicator functions are used to differentiate between existing and non-existing edges and calculate the loss accordingly.

## Claim-guided attention pooling

Building upon the foundation of the GATs, we put forth a modified claim-guided attention mechanism. This mechanism is crafted to operate together with the previously introduced GATs, further refining the representation of our data. Our proposed mechanism enhances both the topical coherence and semantic inference capabilities of the model, ultimately making it more precise in reasoning the veracity of a rumored event.

The claim-guided attention pooling is predicated on the assumption of relevance between posts and the claim. Many posts, when considered without the context of the claim, exhibit low relevance to the claim. Direct attention pooling in such cases may not effectively aggregate the information from the graph.

### Post-level attention

To address the issue of ineffective aggregation of posts with low relevance to the claim when directly applying weighted graph pooling based on the relevance between posts and

the claim, we employ a gating module (*Lin et al., 2021*). This module is used to infuse each post with claim information, thereby leading to higher attention weights for the posts during subsequent event-level attention computations. This is a departure from previous approaches (*Lin et al., 2021*) that positioned the gating module before every layer of the convolution. Instead, we place this module after the last layer of the convolution. This adjustment not only bolsters the model's ability to enhance topical coherence, but also further addresses the issue of weak relevance during subsequent graph attention pooling.

The operation of our model can be presented as follows:

$$g_{c \to i}^L = sigmoid\left(W_g^L h_i^L + U_g^L h_c^L\right) \tag{21}$$

$$\bar{h}_i^L = g_{c \to i}^L \odot h_c^L + \left(1 - g_{c \to i}^L\right) \odot h_i^L \tag{22}$$

$$\tag{22}$$

In this configuration, $g_{c \to i}^L$ represents the contribution of claim $c$ to target post $i$ in the $L$-th layer, consisting of trainable parameters $W_g^L$ and $U_g^L$. $h_c^L$ represents the hidden state of claim $c$ in the $L$-th layer. $\bar{h}_i^L$ represents the updated hidden state of target post $i$ in the $L$-th layer after incorporating the information from the claim. For simplicity, we have omitted the bias. The $\odot$ symbol denotes the Hadamard product.

### *Constrained event-level attention*

We have made a significant adjustment to our model to address the limitations of the traditional GAT-mean-based model, particularly its potential inability to accurately weight node vectors. This mechanism works in tandem with the post-level attention, aiming to boost the semantic inference capacity of our model, with a focus on accurately capturing the relationship between the posts and the claim.

Our approach simplifies the complex joint representation used in previous models (*Lin et al., 2021*). Instead of employing concatenation, element-wise multiplication, and the absolute element-wise discrepancy between $h_c^L$ and $h_i^L$, which is subsequently processed using a fully-connected (FC) layer and TanH, we employ an add-attention mechanism to measure the similarity between $h_c$ and $h_i$. This allows us to reduce the joint representation to a more manageable form:

$$h_i^c = \left[h_c^L \| \bar{h}_i^L\right] \tag{23}$$

This modification, while seemingly simple, provides performance enhancements. We empirically find that it allows the model to effectively process the relationships between posts and the claim, while providing computational efficiencies. In this setup, $\beta_i$ represents the attention weight assigned to post $\mathbf{x}_i$ to obtain the aggregated representation $\hat{s}$ of the event. Building on this, we proceed to implement graph attention pooling on the claim-enhanced post representations. This is done to select informative posts based on inference, guided by the joint representation $h_i^c$. This leads to:

$$b_i = \tanh\left(FC\left(h_i^c\right)\right) \tag{24}$$

$$\beta_i = \frac{\exp(b_i)}{\sum_i \exp(b_i)} \tag{25}$$

$$\hat{s} = \sum_i \beta_i \tilde{h}_i^L \tag{26}$$

Recognizing that not all posts contribute equally to the claim, we refine our model with a thresholding step to exclude less relevant posts. Only posts with attention weights $\beta_i$ exceeding a predefined threshold $q$ are considered, ensuring that our model focuses on the most informative content when constructing the event representation $\hat{s}$.

$$\beta_i = \frac{\exp(b_i)}{\sum_i \exp(b_i)} (\beta_i > q) \tag{27}$$

This constrained event-level attention mechanism significantly refines the information filtering and processing capabilities at the graph level, which is crucial for effective rumor detection. By setting a specific threshold $q$, the model efficiently excludes posts that do not significantly relate to the core topic of the event, thereby ensuring that the graph representation is concentrated on the most critical information.

### Classification of rumor detection

Finally, we use the refined representations obtained from the GAT and the claim-guided attention pooling to classify the rumors. The prediction result $\hat{y}$ of the event is calculated by a multi-layer perceptron(MLP).

$$\hat{y} = softmax(MLP(\hat{s})). \tag{28}$$

Then, we employ the cross-entropy loss and relation-guided attention loss to train our model using the ground truth $y$:

$$\mathcal{L}(y, \hat{y}) = -\sum_{i=1}^{N} y_i log(\hat{y}_i) + \lambda \mathcal{L}_E^l. \tag{29}$$

In this loss function $\mathcal{L}$, $\lambda$ balances the cross-entropy loss and relation-guided attention loss.

We provide pseudo-code for our training procedure in Algorithm 1.

## EXPERIMENTS AND ANALYSIS

To examine the efficacy of our introduced SAMGAT model, we compare the rumor detection performance with the state-of-the-art models on public datasets. We also compare and analyze the performance of the SAMGAT model and the baseline method in early rumor detection.

---

**Algorithm 1** The SAMGAT Algorithm

---

    **Input:** Graph $G(V, E)$, Event set $C$, Label set $Y = N, F, T, U$

    **Output:** Classifier $f : C \rightarrow Y$

    **for** each layer $l = 1$ to $L$ **do**

4:        **for all** nodes $i \in V$ **do**

            **for all** neighbors $j \in \mathcal{N}_i$ **do**

                Compute unnormalized attention scores using Eq. (5), Eq. (6) and Eq. (11)

                Apply dropout to modify unnormalized attention scores with probability $d$

8:            **end for**

            Apply softmax to unnormalized attention scores, obtaining attention scores

            Apply TopK to attention scores

            Update node representations using Eq.(16)

12:        **end for**

    **end for**

    Apply post-level attention for each post $i$ using Eq. (21) and (22)

    Compute event-level representation using Eq. (23) to (27)

16: Make predictions $\hat{y}$ with classifier and softmax using Eq. (28)

    Compute loss $\mathcal{L}(y, \hat{y})$ and $\mathcal{L}_E$ to update model parameters using Eq. (20) and Eq.(29)

---

## Experimental settings
### Datasets

We evaluate our proposed method on three publicly available benchmark datasets: Twitter15, Twitter16 (*Ma, Gao & Wong, 2017*), and PHEME (*Zubiaga, Liakata & Procter, 2016*). These datasets have been widely adopted by the research community and are considered standard benchmarks for evaluating rumor detection models. By utilizing the same datasets as previous studies, we can directly compare the performance of our proposed SAMGAT model with state-of-the-art approaches, ensuring the consistency and reliability of our experimental results. For the text information of the graph nodes, we employ the Bidirectional Encoder Representations from Transformers (*Nguyen, Vu & Tuan Nguyen, 2020*), which has demonstrated effectiveness in areas including Sentiment Analysis (*Zhang et al., 2020*), natural language inference, *etc.*, to encode every post's content to form feature matrix X. The statistics are presented in Table 2. The claims in Twitter15 and Twitter16 are labeled as non-rumor, false rumor, true rumor, or unverified rumor, while the unbalanced dataset PHEME contains two binary labels: false rumor and non-rumor.

The labels for each event in Twitter15 and Twitter16 were annotated based on the veracity tag (*Ma, Gao & Wong, 2017*) of the corresponding item on rumor-dispelling websites such as snopes.com and Emergent.info. The labels of the PHEME dataset are annotated by news practitioner.

### Compared methods

We evaluate our proposed model against the following state-of-the-art baselines:

**Table 2  Statistics of the datasets.**

| Statistic | Twitter15 | Twitter16 | PHEME |
|---|---|---|---|
| # of source tweets/events | 1490 | 818 | 6425 |
| # of non-rumors | 374 | 205 | 4023 |
| # of false rumors | 370 | 205 | 2402 |
| # of unverified rumors | 374 | 203 | – |
| # of true rumors | 372 | 205 | – |
| # of users | 276,663 | 173,487 | 48,843 |
| # of posts/postings | 331,612 | 204,820 | 197,852 |

- DTC (*Castillo, Mendoza & Poblete, 2011*): A decision tree based approach that combines various news features.
- SVM-TS (*Ma et al., 2015*): A support vector machine classifier that models the temporal properties of social context during message propagation.
- GRU-RNN (*Ma et al., 2016*): This article introduces an RNN based model to detect rumors, which learns from the temporal dynamics of social media to identify rumors more effectively than methods relying on static features.
- BU-RVNN and TD-RVNN (*Ma, Gao & Wong, 2018*): Models that view rumor propagation as a tree structure and adopt bottom-up and top-down recursive neural networks for the rumor classification task.
- PLAN (*Khoo et al., 2020*): A tree transformer based model capturing long-term interactions with token-level and post-level attention.
- BiGCN (*Bian et al., 2020*): An approach using bidirectional graph convolutional models for social media rumor detection that analyzes both propagation and dispersion patterns.
- ClaHi-GAT (*Lin et al., 2021*): A GAT model based on an undirected graph, which employs claims to enhance reply posts and incorporate sibling connections between relevant messages.
- HDGCN (*Yu et al., 2022*): An approach to dynamic rumor detection using heterogeneous graph convolutional networks and an ordinary differential equation system.
- TISN (*Luo et al., 2022*): This study combines text and propagation structure by employing BERT and GCN. TISN arranged tweets in chronological order to extract temporal features of rumors.

In line with standard evaluation practices within the field, we assessed model performance using accuracy and F1 score to offer a comprehensive perspective on model performance, considering both precision and recall. We employed 5-fold cross-validation to ensure that our assessment is robust and reliable, providing a thorough validation across various subsets of data. This approach strengthens the credibility of our findings by demonstrating consistent performance across different partitions of the dataset.

The BERT word embedding is initialized with the size of 768. We configured the model with 2 graph attention layers (represented by $L$) and 8 attention heads (notated as $K$). The hyperparameters are set as follows: $p_n = 0.5$, $r_c = 0.9$, $r_n = 0.9$, $r_s = 0.4$,

**Table 3  Results on the Twitter15 dataset.**

| Method | ACC | F1 | | | |
|---|---|---|---|---|---|
| | | NR | FR | TR | UR |
| DTC | 0.454 | 0.733 | 0.355 | 0.317 | 0.415 |
| SVM-TS | 0.544 | 0.796 | 0.472 | 0.404 | 0.484 |
| BU-RVNN | 0.708 | 0.695 | 0.728 | 0.759 | 0.653 |
| TD-RVNN | 0.723 | 0.682 | 0.758 | 0.821 | 0.654 |
| BiGCN | 0.836 | 0.791 | 0.842 | 0.887 | 0.801 |
| ClaHi-GAT | 0.891 | 0.878 | 0.882 | **0.931** | 0.867 |
| TISN | 0.886 | **0.957** | **0.893** | 0.869 | 0.812 |
| HDGCN | 0.834 | 0.853 | 0.868 | 0.859 | 0.823 |
| **SAMGAT** | **0.917** | 0.928 | 0.862 | 0.908 | **0.893** |

**Notes.**
The bold value indicates the best result among all methods.

$p_a = 5$, $y^s = 0.4$, $q = 0.005$, and $\lambda = 1$. Parameters were updated *via* backpropagation using the Adam optimizer (*Kingma & Ba, 2015*), and the initial learning rate is 0.0005. Dropout regularization of 0.2 was applied to prevent overfitting. Early stopping (*Yao, Rosasco & Caponnetto, 2007*) was employed to monitor validation loss, stopping training if loss did not improve for 10 epochs in order to avoid overfitting. The code has been published at https://github.com/qwerdabc/SAMGAT. The dataset link for Twitter15 and Twitter16 can be found at https://github.com/majingCUHK/Rumor_RvNN, the original data source by *Ma, Gao & Wong (2018)*. For the PHEME dataset, it is available at https://doi.org/10.6084/m9.figshare.4010619.v1.

## Results and analysis

In this section, we assess the effectiveness of our proposed method for identifying rumors. The results of our method and all baseline methods, as presented in Tables 3, 4 and 5, indicate that our SAMGAT outperforms the existing baselines across the majority of evaluation metrics, even in the case where data is unbalanced. The reasons for SAMGAT's superior performance can be attributed to the following factors:

Feature-based approaches like SVM-TS and DTC struggle to perform well because they rely on manually engineered features derived from overall tweet statistics. These features alone are not enough to fully characterize the general properties of tweets and model the complex dynamics of how information spreads. SVM-TS achieves marginally better performance due to its use of a comprehensive feature set and its focus on the temporal patterns of retweets. However, it still falls short of capturing the intricate relationships and dependencies within the data.

SAMGAT outperforms the two recursive neural network (RvNN) models introduced by *Ma, Gao & Wong (2018)*. Key reasons for this include the earlier development of RvNN, possibly resulting in a less advanced model capacity. Moreover, RvNN models tend to lose track of past data during propagation, a significant drawback in rumor detection where the original tweet's history is vital. This issue diminishes the importance of source

**Table 4  Results on the Twitter16 dataset.**

| Method | ACC | F1 | | | |
|---|---|---|---|---|---|
| | | NR | FR | TR | UR |
| DTC | 0.465 | 0.643 | 0.393 | 0.419 | 0.403 |
| SVM-TS | 0.574 | 0.755 | 0.420 | 0.571 | 0.526 |
| BU-RVNN | 0.718 | 0.723 | 0.712 | 0.779 | 0.659 |
| TD-RVNN | 0.737 | 0.662 | 0.743 | 0.835 | 0.708 |
| BiGCN | 0.864 | 0.788 | 0.859 | 0.932 | 0.864 |
| ClaHi-GAT | 0.908 | 0.862 | 0.916 | **0.954** | **0.901** |
| HDGCN | 0.865 | 0.820 | 0.863 | 0.930 | 0.863 |
| TISN | 0.883 | **0.956** | 0.797 | 0.802 | 0.833 |
| **SAMGAT** | **0.913** | 0.932 | **0.921** | 0.940 | 0.838 |

**Notes.**
The bold value indicates the best result among all methods.

**Table 5  Results on the PHEME dataset.**

| Method | Acc. | F1 Score | |
|---|---|---|---|
| | | Non-rumor | Rumor |
| DTC | 0.670 | 0.755 | 0.494 |
| SVM-TS | 0.685 | 0.757 | 0.539 |
| GRU-RNN | 0.775 | 0.832 | 0.658 |
| RvNN | 0.829 | 0.873 | 0.736 |
| PLAN | 0.824 | 0.868 | 0.731 |
| Bi-GCN | 0.835 | 0.872 | 0.764 |
| ClaHi-GAT | 0.859 | **0.893** | 0.790 |
| **SAMGAT** | **0.864** | 0.865 | **0.863** |

**Notes.**
The bold value indicates the best result among all methods.

tweets in RvNN models. In contrast, the graph model's input encompasses the attributes of all nodes, unlike RvNN's moment-specific input. Additionally, the graph network's propagation is guided by edge relationships, as opposed to RvNN's dependency on the sequence of reading. These factors contribute to RvNN's suboptimal performance in this context.

BiGCN, HDGCN, and TISN employ graph convolutional networks to simulate rumor propagation, integrating structural characteristics for detecting rumors. GCN highlights node connections and synchronizes feature sharing across nodes. Unlike the RNN recursive model, the graph model processes the entire graph in a single pass, leading to a global and potentially more stable representation. However, the method based on graph convolutional neural networks cannot dynamically assign weights and reduce the influence of noisy nodes. BiGCN adopts a bi-directional modeling approach for rumor propagation graphs, utilizing top-down and bottom-up graph convolutional networks

to extract representations. However, this bi-directional propagation process has two key limitations. First, it propagates and amplifies information from noisy nodes twice. Second, it treats all relationships within the graph as one, failing to differentiate between more and less significant connections. TISN concatenates all reply posts in chronological order to capture temporal information and uses GCN to capture propagation structure information. Yet, its approach to temporal and structural modeling does not effectively filter out irrelevant posts and lacks sufficient utilization of the information from the original posts, making it inferior to SAMGAT in this regard. HDGCN employs an ODE-based GCN to capture the dynamic propagation of messages, which can be viewed as the aggregation of results from multiple GCN layers with smaller propagation steps. Although this approach provides more fine-grained modeling compared to discrete methods, the propagation process still includes noisy nodes. Additionally, it employs mean-pooling to aggregate graph-level representations, which does not distinguish contributions from different nodes, leading to lossy modeling.

ClaHi-GAT and SAMGAT both model conversational structures using undirected interaction graphs, aiming to improve representation learning by considering comprehensive social contexts and focusing on semantically relevant posts related to the target claim. However, they differ in several key aspects. At the node aggregation level, ClaHi-GAT employs the original GAT formula, which struggles to effectively differentiate the importance of diverse interactions. In contrast, SAMGAT utilizes GATv2 and dot-product attention mechanisms, enabling the model to selectively weaken or discard less relevant connections, refining its ability to focus on significant interactions within the network. Moreover, SAMGAT introduces a top-k mechanism for node selection and weighted multi-head attention to differentiate the importance of various relationships, further enhancing its capacity to prioritize pertinent information and filter out noise. In terms of graph aggregation, both models adopt claim-guided attention to weight and aggregate nodes. However, SAMGAT takes it a step further by incorporating a threshold to filter out nodes with low attention scores relative to the claim, ensuring that only the most pertinent information is considered during the graph readout process. This aligns with SAMGAT's node-level constraint strategy, jointly elevating the model's performance in rumor detection tasks. In comparison, while ClaHi-GAT also distinguishes the importance of different posts through attention mechanisms, it does not impose strict conditions during aggregation, potentially allowing redundant or irrelevant information to influence the final representation. Furthermore, SAMGAT distinguishes itself from ClaHi-GAT by incorporating self-supervised learning techniques to explicitly learn node representations capable of reconstructing the graph structure. This aligns with the goal of rumor detection research considering propagation structures: to effectively leverage graph structure for enhanced performance, ultimately contributing to SAMGAT's superior performance in the rumor detection task.

## Ablation experiments

To validate the contribution of each component in SAMGAT, we make comparisons between the complete model and these derivative versions: SAMGAT-ADD employs

**Table 6   Ablation experiment on the Twitter15, Twitter16 and PHEME.**

| Model | Twitter15 | Twitter16 | PHEME |
|---|---|---|---|
| | Acc. | Acc. | Acc. |
| SAMGAT | 0.917 | 0.913 | 0.864 |
| SAMGAT-ADD | 0.908 | 0.891 | 0.862 |
| SAMGAT-NR | 0.884 | 0.885 | 0.826 |
| SAMGAT-NCLA | 0.906 | 0.907 | 0.859 |
| SAMGAT-NCAP | 0.898 | 0.901 | 0.841 |

solely the original GAT formula without weighted multi-head attention or DynamicGAT attention. SAMGAT-NR removes the relation-guided attention module, losing the capability to optimize attention weights based on structural information in the network. SAMGAT-NCLA removes the constrained local attention module, making the model unable to filter unrelated nodes and lose cardinality information, reducing the quality of representations. SAMGAT-NCAP removes the claim-guided attention pooling module. As a result, the model can no longer aggregate information based on claim cues.

The experimental results are summarized in Table 6. We can observe that: (1) In comparison to SAMGAT, the accuracy of SAMGAT-NR on the Twitter15, Twitter16, and PHEME datasets decreases by 3.3%, 2.8%, and 3.8%, respectively. Clearly, the model's capability is confined without relation-guided attention due to the lack of structural supervision. (2) The lack of claim-guided attention pooling will impair SAMGAT's performance. As the model aggregates all information without selection, noisy information disrupts the model. (3) The SAMGAT model that introduces DynamicGAT attention and weighted multi-head is better than SAMGAT-ADD, demonstrating the effectiveness of these modifications. (4) When constrained local attention is removed, center node aggregates all neighbors to form its own representation, which will contain irrelevant information and reduce its own information.

### Early rumor detection

Early and accurate rumor debunking is crucial to mitigate their spread and negative impacts. We compare the performance of various detection methods at different stages of rumor propagation, measured by the number of responding posts. We evaluate the methods' accuracy as we progressively analyze validation data in chronological order, stopping once the desired quantity of responses is reached.

Figure 5 illustrates the performances of SAMGAT, Clahi-GAT, Bi-GCN, and RvNN across different deadlines. It is observed that models viewing the propagation as a graph (*e.g.*, SAMGAT, Clahi-GAT, and Bi-GCN) attain superior performance in the initial stages of rumor propagation. Interestingly, the initial performance of the other graph-based models (*e.g.*, ClahiGAT) exhibited some noticeable fluctuations. We speculate this is due to the increasing complexity and noise associated with claim propagation. In contrast, our SAMGAT method demonstrates insensitivity to data variations, resulting in improved stability and robustness. Clahi-GAT and Bi-GCN achieve their saturated performance after approximately 30 posts on Twitter15 and Twitter16, whereas our method continues to

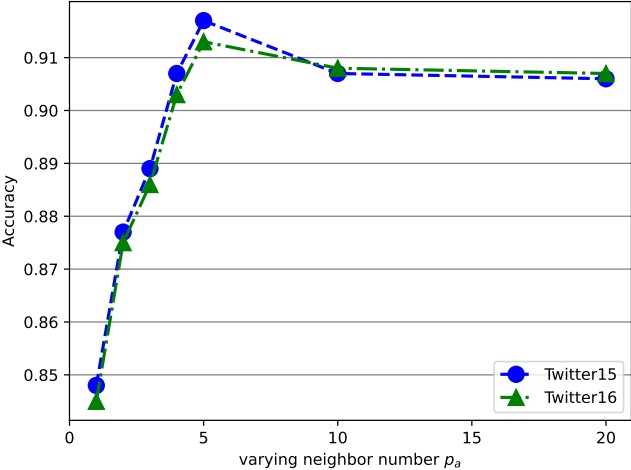

**Figure 5  Early rumor detection accuracy on Twitter15 and Twitter16 datasets.**

improve as the number of posts increases, demonstrating its ability to handle larger and more complex datasets, showcasing a significantly superior early detection performance.

### Sensitive analysis of neighbor number

We also analyze the sensitivity of the hyper-parameter $p_a$ as illustrated in Fig. 6, which controls the aggregation step based on attention weights. Experiments on the Twitter15 and Twitter16 datasets were conducted using constrained local attention by varying $p_a$ in the range of 1,2,3,4,5,10,20. The results show that the accuracy first increases to a peak value and then stabilizes or slightly decreases. This indicates that $p_a$ plays a crucial role in regulating the balance between preserving relevant information and filtering out noisy or irrelevant nodes. When $p_a$ is too small, the model may not sufficiently aggregate relevant information from neighboring nodes, leading to suboptimal performance. On the other hand, when $p_a$ is too large, the model may include more irrelevant or noisy nodes in the aggregation, which can introduce noise into the graph representation and degrade the rumor detection performance. In the context of rumor detection tasks, the optimal value of $p_a$ allows the model to preserve the most relevant nodes to the central node while discarding irrelevant ones, which helps eliminate noisy edges in the propagation tree and filter out off-topic responses.

### CONCLUSION

The proposed SAMGAT model has demonstrated superior performance in rumor detection on social networks, showcasing its ability to navigate the complexities inherent in social media data. The incorporation of DynamicGAT allows the model to discern the importance of various relations within the graph, improving the model's robustness to graph noise. The introduction of a self-supervised task and a dual claim-guided mechanism further refine the graph and maintain cardinality information, thereby contributing to the efficacy of rumor detection. The model's ability to effectively detect rumors and prioritize pertinent

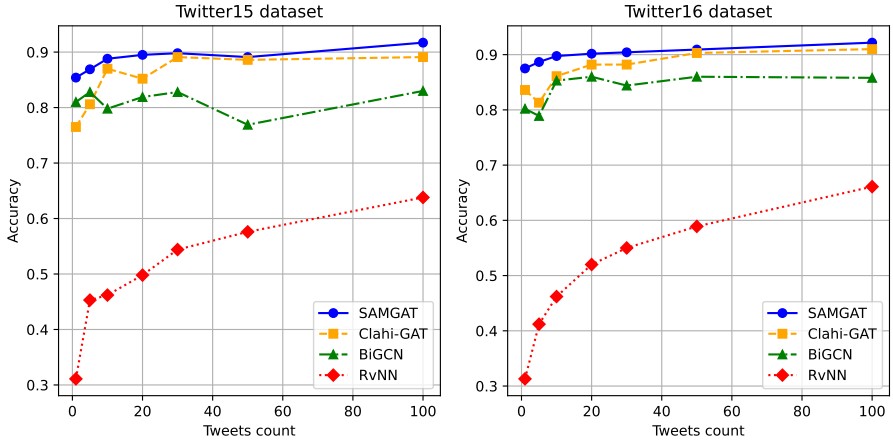

**Figure 6** Impact of number of neighbors $p_a$ on rumor detection accuracy.

information can contribute to the development of more robust systems for combating the spread of misinformation on social media platforms. Moreover, SAMGAT can be integrated into real-time rumor detection pipelines and content moderation systems to identify and address the spread of misinformation in a timely manner. Social media platforms and news organizations can leverage the model's capabilities to enhance their content verification processes. Future work could explore the application of the proposed model in other domains, such as fake news detection or stance classification. Additionally, our study is currently limited to event-level rumor detection. However, as multiple events often pertain to the same topic, incorporating topic-based analysis could prove beneficial, especially for events characterized by a limited volume of posts. Exploring this avenue may enhance the robustness and scope of our rumor detection methodology in future research.

### Funding
This work was supported by the National Natural Science Foundation of China under grant 62006009. The funders had no role in study design, data collection and analysis, decision to publish, or preparation of the manuscript.

### Grant Disclosures
The following grant information was disclosed by the authors:
The National Natural Science Foundation of China: 62006009.

### Competing Interests
The authors declare there are no competing interests.

### Author Contributions
- Yafang Li conceived and designed the experiments, analyzed the data, authored or reviewed drafts of the article, and approved the final draft.

- Zhihua Chu conceived and designed the experiments, performed the experiments, analyzed the data, performed the computation work, prepared figures and/or tables, authored or reviewed drafts of the article, and approved the final draft.
- Caiyan Jia analyzed the data, authored or reviewed drafts of the article, and approved the final draft.
- Baokai Zu conceived and designed the experiments, authored or reviewed drafts of the article, and approved the final draft.

### Data Availability

The code is available at GitHub and Zenodo:

- https://github.com/qwerdabc/SAMGAT

- qwerdabc. (2024). qwerdabc/SAMGAT: init (init). Zenodo. https://doi.org/10.5281/zenodo.12598037.

The data for Twitter15 and Twitter16 is available at Github and figshare:

- https://github.com/majingCUHK/Rumor_RvNN

- ma, jing (2017). rumdetect2017. figshare. Dataset. https://doi.org/10.6084/m9.figshare.25406389.v1

The data for PHEME is available at figshare:

Zubiaga, Arkaitz; Wong Sak Hoi, Geraldine; Liakata, Maria; Procter, Rob (2016). PHEME dataset of rumours and non-rumours. figshare. Dataset. https://doi.org/10.6084/m9.figshare.4010619.v1.

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
