# Peer review of "SAMGAT: structure-aware multilevel graph attention networks for automatic rumor detection"

_PeerJ Computer Science, doi:10.7717/peerj-cs.2200_

## Round 0.1 · original submission · Major Revisions

According to the reviewers, the manuscript must be revised accordingly.

Reviewer 1 ·

Basic reporting

The manuscript entitled “SAMGAT: Structure-aware multilevel graph attention networks for automatic rumor detection” has been investigated in detail. The paper introduces SAMGAT, a novel Structure-Aware Multilevel Graph Attention Network, for rumor classification on social platforms like Twitter. SAMGAT leverages a dynamic attention mechanism to filter and process relevant information, improving rumor detection efficacy. While claiming superior performance compared to existing methods, the paper lacks detailed methodology explanation and empirical evaluation. Clearer descriptions and comprehensive experimental results are needed to substantiate the proposed approach's effectiveness in combating misinformation spread. There are some points that need further clarification and improvement:
1) The problem of misinformation dissemination on social platforms is indeed critical, but the introduction lacks a comprehensive discussion on the nuances of this issue, such as the impact of fake news on society, the challenges of rumor detection, and the limitations of existing methods. Providing this context would better frame the significance of the proposed approach.

Experimental design

2) While the SAMGAT model is introduced as a novel approach, the explanation of its architecture and functioning is vague. The description lacks clarity on how exactly the dynamic, structure-aware attention mechanism operates and how it improves information filtering and processing. Providing more detailed insights into the model's inner workings would enhance the understanding and reproducibility of the proposed method.

3) The empirical evaluation appears to be superficial, with limited details on the experimental setup, including the selection of benchmark datasets, evaluation metrics, and comparison with state-of-the-art methods. Without a thorough analysis of these aspects, it is challenging to assess the true effectiveness and generalizability of SAMGAT.

Validity of the findings

4) While the paper claims that SAMGAT outperforms existing methods, the lack of detailed experimental results and analysis weakens this assertion. Additionally, the discussion on the broader implications and potential applications of SAMGAT in mitigating the spread of misinformation is limited.

Overall, the paper addresses an important issue but falls short in providing a clear and detailed explanation of the proposed method and its empirical validation. Enhancing the clarity of the methodology description, providing comprehensive experimental results, and discussing the broader implications of the findings would significantly strengthen the paper's contribution to the field of rumor detection in social media.

Additional comments

The manuscript entitled “SAMGAT: Structure-aware multilevel graph attention networks for automatic rumor detection” has been investigated in detail. The paper introduces SAMGAT, a novel Structure-Aware Multilevel Graph Attention Network, for rumor classification on social platforms like Twitter. SAMGAT leverages a dynamic attention mechanism to filter and process relevant information, improving rumor detection efficacy. While claiming superior performance compared to existing methods, the paper lacks detailed methodology explanation and empirical evaluation. Clearer descriptions and comprehensive experimental results are needed to substantiate the proposed approach's effectiveness in combating misinformation spread. There are some points that need further clarification and improvement:
1) The problem of misinformation dissemination on social platforms is indeed critical, but the introduction lacks a comprehensive discussion on the nuances of this issue, such as the impact of fake news on society, the challenges of rumor detection, and the limitations of existing methods. Providing this context would better frame the significance of the proposed approach.
2) While the SAMGAT model is introduced as a novel approach, the explanation of its architecture and functioning is vague. The description lacks clarity on how exactly the dynamic, structure-aware attention mechanism operates and how it improves information filtering and processing. Providing more detailed insights into the model's inner workings would enhance the understanding and reproducibility of the proposed method.
3) The empirical evaluation appears to be superficial, with limited details on the experimental setup, including the selection of benchmark datasets, evaluation metrics, and comparison with state-of-the-art methods. Without a thorough analysis of these aspects, it is challenging to assess the true effectiveness and generalizability of SAMGAT.
4) While the paper claims that SAMGAT outperforms existing methods, the lack of detailed experimental results and analysis weakens this assertion. Additionally, the discussion on the broader implications and potential applications of SAMGAT in mitigating the spread of misinformation is limited.
Overall, the paper addresses an important issue but falls short in providing a clear and detailed explanation of the proposed method and its empirical validation. Enhancing the clarity of the methodology description, providing comprehensive experimental results, and discussing the broader implications of the findings would significantly strengthen the paper's contribution to the field of rumor detection in social media.

Reviewer 2 ·

Basic reporting

no comment

Experimental design

no comment

Validity of the findings

no comment

Additional comments

The article is devoted to the urgent problem of spreading rumors through social networks, in which we can find a description of methods for classifying and early detecting rumors using graph and neural network methods. However, this study raises some questions about the analysis of only sets of source materials from one social network, Twitter, thus questioning the possibility of applying the developed method to other networks, due to the different data set in the analysis. The analysis of international experience is also based on the research of geographically related authors, which does not allow us to claim a qualitative meta-analysis of the problem and the choice of the most effective method for experimental work. In addition, the models do not have a clear correlation between ordinary rumors and those generated by opinion leaders on social media and do not properly analyze the speed of information dissemination in correlation with the number of followers of a particular entity. It would also be interesting to consider methods of analyzing the use of artificial intelligence (bots) to spread rumors from fake accounts. In general, assessing the article positively, I believe that, subject to the relevant recommended changes, it can be recommended for publication in the journal.

Reviewer 3 ·

Basic reporting

The authors proposed a multi-level graph detection-based rumor detection approach. While the article appears interesting and novel, it requires major revisions.
The abstract is poorly written as it fails to accurately reflect the contributions. Furthermore, it only offers generic discussions rather than presenting actual contributions.
How are Structure-Aware Multilevel Graph Attention Networks (SAMGAT) for rumor classification superior to benchmarks?
This needs to be clearly mentioned in the abstract and results section. Moreover, there should be a clear understanding of improvement reasons in the results.

The related work needs more recent studies and a clearer, more in-depth discussion to attract the interest of readers.
All equations should be numbered, and each mathematical equation should be presented with detailed discussions, including all symbols.

The design of Figure 1 is appreciated; it clearly illustrates the functioning of the complete framework.
However, the symbols should be checked again, as many of them are overlapping and not defined.

The results are well-presented, but they need a more in-depth discussion of the superiority of this algorithm.

Experimental design

Check Basic Reporting.

Validity of the findings

Check Basic Reporting.

---

## Round 0.2 · accepted · Accept

Due to the unavailability of one reviewer, I personally reviewed the comments and the manuscript can be accepted.

Reviewer 1 ·

Basic reporting

My comments have been addressed. It is acceptable in the present form.

Experimental design

My comments have been addressed. It is acceptable in the present form.

Validity of the findings

My comments have been addressed. It is acceptable in the present form.

Additional comments

My comments have been addressed. It is acceptable in the present form.

Reviewer 3 ·

Basic reporting

I have carefully examined the revised paper. The authors have successfully addressed each comment. I would recommend the acceptance of the article.

Experimental design

No comment

Validity of the findings

No comment